# Approach of Pregnant Women from Poland and the Ukraine to COVID-19 Vaccination—The Role of Medical Consultation

**DOI:** 10.3390/vaccines10020255

**Published:** 2022-02-08

**Authors:** Sławomir Januszek, Natalia Siwiec, Rafał Januszek, Marta Kluz, Roman Lebed, Paweł Toś, Tomasz Góra, Krzysztof Plens, Krzysztof Dąbrowski, Marcin Sidorowicz, Aleksandra Szcześniewska, Edyta Barnaś, Katarzyna Kalandyk-Osinko, Dorota Darmochwal-Kolarz, Tomasz Kluz

**Affiliations:** 1Department of Gynecology, Oncology and Obstetrics, Fryderyk Chopin University Hospital No. 1, 35-055 Rzeszów, Poland; natalia_s@opoczta.pl (N.S.); kalandyk@op.pl (K.K.-O.); jtkluz@interia.pl (T.K.); 2Department of Gynecology and Obstetrics, Institute of Medical Sciences, Medical College of Rzeszów University, 35-316 Rzeszów, Poland; ebarnas@interia.eu (E.B.); dorotak@mp.pl (D.D.-K.); 3Department of Cardiology and Cardiovascular Interventions, University Hospital, 30-688 Kraków, Poland; jaanraf@interia.pl; 4Department of Pathology, Fryderyk Chopin University Hospital No. 1, 35-055 Rzeszów, Poland; marta.kluz@interia.pl; 5Khmelnytsky Regional Perinatal Centre, 29-016 Khmelnytskyi, Ukraine; roman_lebed@ukr.net; 6Department of Gynecology and Obstetrics, University Hospital No. 2, 35-301 Rzeszów, Poland; pwtos@wp.pl; 7Department of Gynecology and Obstetrics, Jan Paweł II Hospital, 35-241 Rzeszów, Poland; tomasz.gora.1983@wp.pl; 8Department of Obstetrics and Perinatology, Medical College, Jagiellonian University, 31-501 Kraków, Poland; 9KCRI, 30-055 Kraków, Poland; plens_krzysztof@o2.pl; 10Department of Perinatology, City Hospital in Ruda Śląska, Gynecology and Obstetrics, 41-717 Ruda Śląska, Poland; lekarz00@interia.pl (K.D.); asisik@wp.pl (M.S.); 11Departament of Perinatology and Gynecology, Polish Mother’s Memorial Hospital Research Institute, 93-338 Łodź, Poland; aleksandra@szczesniewski.pl

**Keywords:** COVID-19 vaccination, pregnancy, medical counselling, acceptance, hesitancy, attitude, intention to vaccinate

## Abstract

There are many arguments for the safety and efficacy of COVID-19 vaccines in pregnancy. The aim of this study is to describe the level of vaccination acceptance, to find the factors that most influence the decision to vaccinate, and to describe the scale of changes in vaccination acceptance influenced by medical information on the safety, efficacy, and benefits of vaccination among pregnant women. A total of 300 patients completed the questionnaire, including 150 in Poland and 150 in the Ukraine. The level of vaccination acceptance was assessed before and after medical consultation. There were 53 (35.3%) patients with the intention to get vaccinated in Poland and 25 (16.7%) in the Ukraine. After consultation with a physician, this increased to 109 (72.6%) in Poland and 69 (46%) in the Ukraine. The main factors influencing the acceptance of vaccinations were the fear of harming the foetus (OR-0.119, CI-0.039–0.324 *p* < 0.001), complications in pregnancy (OR-0.073 CI-0.023–0.197 *p* < 0.001), and limitations in the vaccination programme (OR-0.026 CI-0.001–0.207 *p* < 0.001). Medical information about the safety, effectiveness and benefits of vaccinations among pregnant women, provided during a medical visit, may increase the acceptance of vaccinations by 105.6%, as among Polish patients, and by 176%, as among pregnant women from the Ukraine.

## 1. Introduction

Massive loss of human life and health due to the COVID-19 pandemic has created challenges for healthcare systems, destroying supply chains as well as the economy, triggering a mental health crisis [1,2,3]. In a systematic review of 97 studies by Gómez-Ochoa et al. [4], it was estimated that the incidence of SARS-CoV-2 infection from healthcare workers samples was 11% using reverse transcription polymerase chain reaction (95% confidence interval (CI): 7, 15), and 7% (95% CI: 4, 11) using the presence of antibodies [4]. The COVID-19 pandemic has a negative impact on the mental and physical health of healthcare workers [5,6], and particularly unfavorable circumstances leads to severe disease and death [4,7]. The concept of severe disease includes a course that requires hospitalisation, intensive therapy, respiratory support, highly specialised respiratory equipment, or a fatal course of the disease. In some previously published reports, it was verified that in the case of pregnant women, infection with severe acute respiratory syndrome-Coronavirus-2 (SARS-CoV-2) is more frequently connected with severe disease, rendering it necessary to perform ventilation in an invasive manner: extracorporeal membrane oxygenation. In some cases, this may even cause death [8,9,10,11,12]. In a different research trial, an association was demonstrated between COVID-19 infection in pregnancy and the risk of preterm deliveries and caesarean sections [13]. There are also reports indicating vertical transmission with regard to the virus [14]. This can lead to fetal oedema or even death [15]. Among those more prone to asymptomatic infections, we may include children, who further transmit the virus to pregnant women [16,17]. Infection with COVID-19 in pregnancy is a risk factor for preterm birth and other detrimental pregnancy-related outcomes in comparison to women who are pregnant but are not infected with COVID-19 [1]. Vaccines can decrease the pace of an epidemic if vaccination is broadly accepted. A study in southern Italy of 1041 respondents over the age of 65 showed a high percentage (92.7%) of people who were vaccinated and were willing to vaccinate [18]. There are many existing arguments indicating the safety of vaccines for COVID-19. Studies on mRNA vaccines allow us to state that the immunogenicity, safety level, as well as tolerability of these precautionary measures taken among pregnant women do not differ from those applied in non-pregnant women at an analogous age [19]. COVID-19 vaccines comprise mRNAs encapsulated in a lipid nanoparticle, which are further transmitted to cells. The host cells are responsible for the production of coronavirus spike proteins. They stimulate antibody formation. This process is further heightened in regional lymph nodes [20]. COVID-19 vaccination can cause a fever, which in most cases, lasts a maximum of 2 days, but elevated temperatures are not uncommon in pregnancy and can be successfully lowered with acetaminofen. Vaccinated pregnant women experience COVID-19 infection less frequently than their unvaccinated peers, and vaccination for COVID-19 at the time of pregnancy is not connected with more serious pregnancy or delivery-related complications [21]. What is also worth highlighting is that the COVID-19 pandemic has caused many anxieties among pregnant women about both their own health and the health of the foetus, which significantly affects their well-being [22]. In addition, pregnant women may also play an important role in other family members receiving the vaccine. This especially applies to vaccination in childhood. In a systematic review of 16 studies, it was established that attitudes towards pediatric vaccinations indicate, in general, that pregnant women believe vaccines to be important for the protection of their children and the community, but various concerns and misunderstandings persist around vaccine safety and efficacy, which reduce the trust of expectant mothers towards immunization [23]. On the other hand, in an Australian study, it was shown that vaccine decision-making, being discussed and decided prenatally, as well as the quality of delivered information by obstetricians, may reduce vaccine hesitancy among all mothers during pregnancy and post-delivery [24]. Initial acceptance studies on the COVID-19 vaccine predicted an unprecedented challenge regarding its global acceptance [25,26,27]. The Polish Society of Gynaecologists and Obstetricians, based on its own research, observations and published world-data, agrees with the presented positions of ACOG, the Royal College of Obstetricians and Gynaecologists (RCOG), Centres for Disease Control and Prevention (CDC), Society for Maternal-Fetal Medicine (SMFM), stating that pregnant women should be vaccinated against COVID-19 [28]. Women planning pregnancy should be encouraged to complete the vaccination course before conception [28]. The approach to vaccination and predictors of vaccination against COVID-19 among pregnant patients in Poland and Ukraine have not been described in the literature. In studies on vaccination acceptance among pregnant women, predictors of positive vaccination acceptance include: older age, higher level of education, higher income, and access to reliable information on vaccination [1,29,30,31,32]. However, the level of vaccination acceptance in the group of pregnant patients before and after medical consultations during routine medical visits are not described in the literature. There are no studies describing the possible scale of change in the level of acceptance of vaccination against COVID-19 after medical consultation, emphasising the safety of vaccination for the foetus and mother, health benefits of vaccination, vaccination effectiveness, and current guidelines of scientific societies regarding vaccination. The scope of changes in the level of vaccination acceptance in the Polish and Ukrainian populations observed in our study may potentially be analogous in other countries, and become an inspiration to conduct similar studies on larger groups of respondents. The results obtained by demonstrating the value of providing accurate information to patients can impact healthcare professionals and providers and modify the frequency and extent of vaccination consultations.

### Objectives

The aim of this study is to describe the level of vaccination acceptance and to find the factors that most influence the decision to vaccinate among pregnant women from Poland and Ukraine. Another objective of this research is to describe the scale of changes in vaccination acceptance, influenced by medical information on the safety, efficacy, and benefits of vaccination among pregnant women.

## 2. Methodology

### 2.1. Material and Methods 

This cross-sectional study was conducted at the Provincial Clinical Hospital No. 1 in Rzeszów and at the Khmelnytsky Perinatal Perinatal Centre from the beginning of June to the end of August 2021. A total of 300 patients completed the questionnaire, including 150 in Poland and 150 in the Ukraine. Written informed consent was obtained from all participants. The applied protocol was approved by the local ethics committee. The study enrolled consecutive pregnant women who attended routine pregnancy visits, and were given consecutive numbers. The questionnaire consisted of 2 parts: 30 questions in the first part, and 18 in the second part. Both parts of the questionnaire were marked with numbers assigned to the patient. In the first part of the survey, the questions concerned age, place of residence, education, comorbidities, number of births, miscarriages, safety, effectiveness, frequency, severity of side-effects related to vaccinations, current vaccination status, future intention to get vaccinated, and the reason for not accepting the vaccination. The completed first part of the questionnaire was left by the patient in a container intended for this purpose before the visit. Medical consultations were conducted by 11 doctors who specialize in gynaecology or by a specialist gynaecologist as part of routine visits during pregnancy. The number of consultations conducted by individual doctors ranged from 26 to 32. During the visit, the physicians informed patients about the current state of knowledge with regard to the recommendations, safety, effectiveness, and health benefits of vaccination against COVID-19. In the second part of the questionnaire, completed after the medical appointment, the same questions were asked, except for the data that did not change, such as age, number of deliveries, and miscarriages. This part of the questionnaire consisted of 18 questions, and was completed by the patient at home following the medical visit. For the next visit about 3 weeks later, patients brought a completed questionnaire, which they placed in a container intended for this purpose. Then both parts of the questionnaire were paired by the researchers. The patients’ flow charts are presented in Figure 1.

### 2.2. Inclusion Criteria

Pregnant patients who consented to a routine check-up during pregnancy by the doctors of the Provincial Clinical Hospital No. 1 in Rzeszów and the Khmelnytskyi Regional Perinatal Centre were included in the research.

### 2.3. Exclusion Criteria

Patients who did not consent to the study were not included in the research.

### 2.4. Statistical Analysis 

The categorical variables assessed in this study are presented in the form of numbers and percentages. The continuous variables are given as mean ± standard deviation (SD), or as median and interquartile range (IQR). Normality of distribution was evaluated using the Shapiro–Wilk test. Equality of variance was estimated by applying Levene’s test. Group-related differences were subjected to comparison by implementing either Student’s or Welch’s *t*-tests. This was dependent on variance equality for variables demonstrating normal distribution. In the case of continuous variables with non-normal distribution, the Mann–Whitney U-test was applied. The Cochran–Armitage test (for trends) or the Mann–Whitney U-test were implemented for the comparison of ordinal variables. With reference to nominal variables, Pearson’s chi-squared test or Fisher’s exact test (if 20% of cells demonstrated an expected count below 5) were implemented. Uni- and multivariate logistic regression analyses were performed to determine independent predictors of willingness to be vaccinated for COVID-19 before and after consultation with a physician. Due to great amount of considered variables, multivariate models were fitted using forward stepwise regression with the *p* < 0.05 threshold stopping rule. Results of analyses were expressed as odds ratios (OR) along with 95% confidence intervals (95% CI). 

Sample size was determined based on discordant proportions of pairs examined by McNemar’s test [33]. It was assumed that 15% of the pairs would switch from negative to positive attitude towards COVID-19 vaccination and 5% from positive to negative. Additionally, a 25% drop-out rate was assumed. To achieve a power of 90% and two-sided significance of 5% to detect a difference of 10% between the discordant proportions, 288 patients were required. It was decided to slightly increase the sample size to 300 subjects. 

The prepared questionnaire was partially verified. Internal consistency of 18 questions that appeared in the first as well as in the second part of the survey (before and after medical consultation) were examined using McDonald’s omega coefficient. Omega coefficients were equal to 0.93 and 0.91 for measurements before and after medical consultation, respectively, indicating acceptable internal integrity. Although the questionnaire was completed twice by patients, no test–retest reliability indicator could be measured due to different conditions (before and after medical consultation). Validity and discriminatory power of the questionnaire is discussed in the results section (e.g., vaccination attitude differences among patients with different education level in Appendix A).

Two-sided *p*-values < 0.05 were considered statistically significant. Statistical analyses were performed using JMP^®^, Version 16.1.0 (SAS Institute Inc., Cary, NC, USA), and psych package in R: A language and environment for statistical computing, Version 4.1.0 (R Foundation for Statistical Computing, Vienna, Austria) and RStudio: Integrated Development Environment for R, Version 1.4.1717 (RStudio, PBC, Boston, MA, USA).

## 3. Results

### 3.1. Overall Group 

The respondents answered a total of 96% of the questions, while the range of answers to individual survey questions from 1 to 47 ranged from 94–100%. Considering Poland and the Ukraine, before physician consultations, there were 53 (35.3%) patients with the intention to get vaccinated in Poland and 25 (16.7%) in the Ukraine. The difference in the level of vaccination acceptance among Polish and Ukrainian patients before and after medical consultation was statistically significant at the level of *p* < 0.001. After gynaecological consultations, the number of patients who expressed willingness to undergo vaccination increased to 109 (72.6%) in Poland and 69 (46%) in the Ukraine. The value of the effect size was the difference in the level of acceptance after and before medical consultation, and amounted to 37.3% in Poland and 29.3% in Ukraine. This increase in the level of acceptance of vaccinations against COVID-19 after a medical consultation was 105.6% in Poland and 175.4% in Ukraine (Figure 1). 

### 3.2. General Characteristics

In the studied group 16.8% of patients mentioned comorbidities, and 82.4% reported more than three diseases (e.g., obesity, diabetes, hypertension, and hypothyroidism in the same patient). Considering differences between the Polish and Ukrainian population, there were no significant discrepancies except for the frequency of psoriasis and allergies. Differences between the Polish and Ukrainian population related to general characteristics are presented in Appendix A). While considering socioeconomic indices, significant differences were noted in education level, which was higher in Poland, as well as in the number of prior miscarriages, which was greater among Ukrainians (Appendix A). Other differences between the Polish and Ukrainian population concerning the questions included in the questionnaire are presented in Appendix A.

### 3.3. Comparison of Selected Indices and Answers for Questions in the Questionnaire According to the Agreement for Inoculation before and after Consultation with a Gynaecologist

The differences considering general characteristics are presented in Appendix A). Before consultation with a gynaecologist, the patients who accepted vaccination were significantly older. Additionally, patients with hypothyreosis agreed to vaccination against COVID-19 significantly more often before and after physician consultation. The differences in socioeconomic and other questions are presented in Appendix A.

### 3.4. Relationship between Consultation and Approach to Inoculation against COVID-19

After consultation with a gynaecologist, patients were significantly more aware about the severe clinical course of COVID-19 infection in pregnancy (*p* < 0.001). After consultation with a gynaecologist, women more frequently assessed their immunity following vaccination as better compared to the immunity to COVID-19 infection after the disease (*p* < 0.001). More patients after gynaecological consultation were aware that inoculation against COVID-19 is safe during pregnancy (*p* < 0.001). Moreover, they were more convinced about the safety of inoculation against COVID-19 (*p* < 0.001). After gynaecological consultation, fewer patients were afraid of being vaccinated against COVID-19 during pregnancy (*p* < 0.001).

### 3.5. Predictors of Vaccination against COVID-19–Univariate Analysis 

Significant predictors of vaccination against COVID-19 assessed by univariate analysis before and after medical consultation are presented in Appendix A.

### 3.6. Predictors of Vaccination against COVID-19–Multivariate Analysis 

Among the statements significantly related to the decision change associated with the gynaecological consultation, the following could be found: “Are you planning to inoculate your children against COVID-19?”. The more the answer was against vaccination, the lower the probability to change the decision regarding inoculation against COVID-19 after consultation with a gynaecologist. A similar relationship was present for the question: “Do you think the complications after receiving the COVID-19 vaccine are: rare, very rare, common, very common”. The greater the frequency of complications as assessed by the patient, the lower the probability that the decision would change after gynaecological consultation. Additionally, among the predictors of lower vaccination probability, the following were noted: fear of damage to the foetus, fear regarding post-vaccination complications, and limitations in the vaccination programme. Considering the lack of vaccination after consultation with a gynaecologist, significant predictors related to the lack of change in decision were fear of damage to the foetus and post-vaccination complications/adverse events. Predictors assessed by multivariable analysis are presented in Table 1.

### 3.7. Significance of Gynaecological Consultation on Increase in Vaccination Percentage

Statistical significance of the gynaecological consultation on the increase in rate of vaccinations (change in decision to inoculate) among Ukrainian and Polish patients was confirmed by the contingency table and Bowker’s test (Table 2). 

## 4. Discussion

### 4.1. Acceptance of Vaccination—Geographic, Socioeconomic and Medical Aspects

Different levels of vaccination acceptance may be caused by individual, social, or organisational factors. Increased perception regarding infection risk, the benefits of vaccination, restrictions posed by the government, fines for not using masks, as well as intense discussions around the topic of the threat from traditional and/or social media can have a significant influence on people’s willingness to undergo vaccination [34].

Four pregnant women from the Podkarpackie Province and six pregnant women from the Chmielnitsky Province did not agree to participate in the study. Patients who do not consent to participation the study may present a low level of acceptance regarding vaccinations against COVID-19. However, due to their small percentage, this should not significantly change the results of the study. The population of patients enrolled in the study, both in Poland and the Ukraine, is relatively small on the scale of the country and is representative of the population of the studied regions, not the entire country. Before gynaecologist consultations in Poland and the Ukraine, 53 (35.3%) patients in Poland had expressed their willingness to be vaccinated and 25 (16.7%) in Ukraine, which indicated a relatively low level of acceptance among pregnant women compared to research conducted in other countries [1]. Among non-pregnant women in Poland, in an online study conducted by Babicki et al. on 2022 respondents, acceptance of vaccinations in the 18–29 and 30–39 age groups in February 2021 was declared by 49.7 and 29.3%, respectively [35]. In an online survey carried out in March 2021 by Sowa et al. [36], among 885 respondents from the general population in Poland, vaccination acceptance totalled 50.8%. In some sources, it is indicated that the acceptance of vaccinations among the general Ukrainian population is above 50%, noting, at the same time, a large percentage of healthcare workers not accepting vaccinations against COVID-19, i.e., 40% [37,38]. In the studied region of Poland, one of the lowest percentages of people vaccinated against COVID-19 is found, while in the Ukrainian population, the problem of vaccination acceptance is quite common [37,38]. Both before and after gynaecological consultations among pregnant women from the Ukraine, the level of acceptance and vaccinations was significantly lower, which may be due to many reasons. Although the studied populations did not differ significantly from a medical point of view, the level of education of pregnant women from Poland was higher, and the countries of patient origin differ economically, politically, and culturally. The lower acceptance of vaccinations among Ukrainian patients may be due to the non-transparent policy of the Ukrainian authorities regarding the choice of vaccine manufacturer, the availability of the vaccine, as well as the large number of healthcare professionals who do not accept vaccinations [37]. Other authors point to media disinformation, promotion of the antivaccine movement, political distrust, and the issue of purchasing vaccines as the reasons for the low level of acceptance of vaccination against COVID-19 [38]. 

In a study conducted by Skjefte et al. [1] among pregnant women in 19 countries, 52.0% (*n* = 2747) declared the intention to undergo vaccination for COVID-19, while the highest acceptance levels were noted in a study by Tao et al. [39] carried out in China (77.4%), by Mohan et al. [30] in Qatar (75%), and in Italy (74.5%) in a research trial carried out by Mappa et al. [31]. In contrast, among European countries [1], in North America [1,40], Australia [1] as well as Russia [1], lower acceptance levels totalling approximately 50%, were noted. Additionally, in Saudi Arabia, pregnant women or those who are planning pregnancy are more hesitant to receive the COVID-19 vaccination (*p* = 0.001) [41]. In a study by Skiefte et al. [1], the group of respondents was much larger. In many countries, however, it was an online survey. In our study carried out in a face-to-face manner with doctors, we presented the real extent of change in the approach to vaccination under the influence of reliable information provided during a medical visit, which was reassessed at a follow-up visit. In our study, we have shown that this one factor can increase the acceptance of vaccinations by 105.6%, as among patients from Poland, or even by 176%, as among pregnant women from the Ukraine. The increase in acceptance of these vaccinations is significant in both of the analysed countries and it is difficult to directly compare these results with other publications, in which the impact of professional consultations was not assessed prior and post-gynaecologist consultation. Considering socioeconomic indicators, significant differences were found in education, which was higher in Poland, and in the number of previous miscarriages, which was higher among Ukrainians (Appendix A). Before gynaecological consultation, the vaccinated patients were much older. In other studies, older age, higher education, and higher income were also associated with greater acceptance of the vaccine [1,31,42,43]. The proportion of patients with comorbidities was quite high, which may indicate that the selected population is not representative of the pregnant population. However, patients reported comorbidities in the questionnaire by themselves (the questionnaire was anonymous). Some of the diseases mentioned by pregnant women are seasonal allergies or misinterpreted obesity in pregnancy. Many diseases reported by outpatients as disorders could only be part of the diagnostic process, such as blood pressure or glucose monitoring. Among Polish and Ukrainian patients with hypothyroidism, vaccination against COVID-19 was significantly more commonly accepted, which may be related to a higher perceived risk of infection.

### 4.2. Modifiable Factors of Vaccination Acceptance 

Fear of fetal harm, of post-vaccination complications, and limitations in the vaccination programme were also among the predictors of lower vaccination probability in this study. Aspects recurring in other studies concern awareness level regarding COVID-19-related risks, as well as vaccination safety during pregnancy. It is worth noting that in a recent meta-analysis, it was found that the levels of literacy are generally quite low in Europe [44]. Factors such as trusting information received about vaccination [45], being confident of the COVID-19 vaccination safety as well as efficacy [1], believing the importance of vaccination [1], trusting routine childhood vaccines [1], being concerned by the COVID-19 pandemic [1], having trust in public health-agencies, lack of fear related to post-COVID-19 vaccination side side-effects [20], having reliable information [46], explicit communication of information concerning the safety of COVID-19 vaccines among pregnant women [47], having an obstetrician-supervised pregnancy [30], flu vaccination during the previous year [30], and last but not least, having confidence that the COVID-19 vaccine is safe [31] are all factors that seem to have a common denominator. All these aspects concern providing awareness and information to pregnant or breastfeeding women about current knowledge of COVID-19, vaccination against it, or with regard to vaccinations in general. Analogous factors also demonstrate significant effects on the decision to undergo vaccination in other social groups, as well as in the general population [43]. An interesting finding was presented in a study conducted in southern Italy on a population of people over 65 years of age [18]. In this study, a relatively low percentage of respondents declared their willingness to be vaccinated (45.1%) compared to those who were immunized against COVID-19 (86.6%). The authors of the study point out that the decline in acceptance of vaccinations and support for compulsory vaccinations coincided with the introduction of the green pass, which may indicate that compulsory measures should be accompanied by effective education [18].

### 4.3. Gynaecological Counselling

The differences regarding the percentage increase in the number of people willing to vaccinate between Poland and Ukraine may be related to the lower number of patients consulted before the study in the latter country, which further emphasises the role of professional counselling. Communication-based strategies may indicate positive directions for action, including being encouraged by close and trusted individuals, i.e., physicians and/or religious leaders, sharing one’s personal experiences, or being subjected to peer-pressure [47]. It is also worth underlining that evidence-based professional ethics found in the fields of obstetrics and gynaecology provide clear guidelines with regard to vaccination [48,49]. In the study by Cavaliere et al., it has also been confirmed that vaccinations must be recommended to all pregnant women during routine prenatal care, while hesitation before vaccinations can be minimised by consistent recommendation to all pregnant women by medical personnel [50]. This study not only lists professional counselling as one of the factors influencing the acceptance of vaccinations, but also presents a possible increase in acceptance as a result, which has not been described in the literature so far. The scope of changes in the level of vaccination acceptance in the Polish and Ukrainian populations observed in our study may prove to be transferable to populations in other countries, and become an inspiration to conduct similar studies on larger groups of respondents. The results obtained could motivate healthcare providers and health professionals to provide more widespread and targeted medical advice on immunisation.

### 4.4. Limitations

This study is subject to limitations. The study was performed among 300 patients presenting for a routine pregnancy visit to doctors working at clinical centres. Physicians working at clinical centres and patients who come to them for routine check-ups may not completely represent the entire population, in which both knowledge about vaccination and the will to vaccinate may be at different levels. Patients choosing more experienced centres for pregnancy control visits may have a greater burden of comorbidities, and their level of acceptance may be different than in the entire population of pregnant women. Additionally, due to the relatively small group of patients under study, in our analysis we focused on the assessment of indicators and trends that may increase the acceptance of vaccinations. There is no standardized questionnaire examining the level of vaccination acceptance among pregnant women, which is a significant limitation. Therefore, we used the proprietary questionnaire subjecting it to statistical verification.

## 5. Conclusions

The level of acceptance of vaccinations against COVID-19 in the studied group of pregnant women from Poland (35.3%) and Ukraine (16.7%) was relatively low. Concerns about vaccine-related side effects or complications, harm to the foetus, and limitations of the vaccination programme were significant reasons for the low level of vaccination acceptance and important predictors of lower vaccination likelihood. Medical information about the safety, effectiveness, and benefits of vaccinations among pregnant women, provided during a medical visit, may increase the acceptance of vaccinations by 105.6%, as among Polish patients, and even by 176%, as among pregnant women from the Ukraine.

## Figures and Tables

**Figure 1 vaccines-10-00255-f001:**
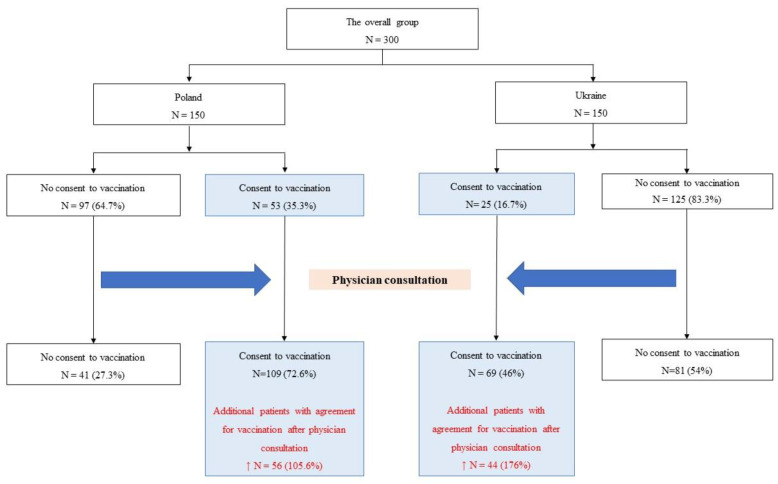
Acceptance of vaccination before and after gynaecologist consultation (Flow chart).

**Table 1 vaccines-10-00255-t001:** Predictors of vaccination against COVID-19 before and after gynaecological consultation assessed by multivariate analysis.

	Before Gynaecological Consultation Odds Ratio (95% Confidence Interval, *p*-Value)	After Gynaecological Consultation Odds Ratio (95% Confidence Interval, *p*-Value)
Are you planning to vaccinate your children against COVID-19? (no vs. yes)	0.277 (0.150–0.477, <0.001)	
Do you think the complications after receiving the COVID-19 vaccine are: (rare/very rare vs. common/very common)	0.406 (0.242–0.642, <0.001)	
Have you been vaccinated against COVID-19?—Reason for negative approach:		
Fear of damage to the foetus (yes vs. no)	0.073 (0.023–0.197, <0.001)	0.024 (0.009–0.057, <0.001)
Fear of post-vaccination complications/adverse reactions (yes vs. no)	0.119 (0.039–0.324, <0.001)	0.040 (0.009–0.057, <0.001)
I have not had such an opportunity yet due to the limitations in the vaccination programme (yes vs. no)	0.026 (0.001–0.20, <0.001)	

**Table 2 vaccines-10-00255-t002:** Contingency table based on the question: Have you been vaccinated against COVID-19? 1. Yes; 2. Yes, before pregnancy; 3. Yes, during pregnancy; 4. The first dose during pregnancy, the second dose I postponed after pregnancy; 5. The first dose before pregnancy—I postponed full vaccination until later in pregnancy; 6. I postponed the first dose until later in pregnancy, full vaccination later in pregnancy or after pregnancy; 7. No 7. No—Whether the patient has been vaccinated against COVID-19 after a gynaecological visit and whether the patient has been vaccinated against COVID-19 before the medical visit.

		Before Medical Visit		
		Yes (1–6)	No (7)	Total
After medicalvisit	Yes (1–6) Count (%)	58 (19.4)	52 (17.4)	110 (36.8)
	No (7) Count (%)	7 (2.34)	182 (60.9)	189 (63.2)
	Total (%)	65 (21.7)	234 (78.3)	299

The *p* value for the Bowker’s test is smaller than 0.0001.

## Data Availability

The data presented in this study are available on request from the corresponding author.

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
