# Peer review of "Approach of Pregnant Women from Poland and the Ukraine to COVID-19 Vaccination—The Role of Medical Consultation"

_vaccines, 2022, doi:10.3390/vaccines10020255_

Round 1

Reviewer 1 Report

Response to the authors

Major points

The authors describe the role of medical consultation in Poland and Ukraine. The represented group of 300 pregnant women is relatively small and it is unclear whether this group is representative for the whole countries or other countries. This has not been discussed.

Pregnant patients who did not consent to the study (page 3, line 116,117) were not included. I suppose that those patients might be against vaccination and might bias the results. Why did they not consent to the survey? This important point has not been discussed.

There is no comparison with a control group of women that are not pregnant in the same age. How is the vaccination readiness in this group or other groups? Why is the focus (returning the society to the pre-pandemic state page 2 line 72) of the authors on vaccination willingness on pregnant women? The focus of the study is unclear.

There is much research on pregnant women in the literature, and factors having the greatest impact for refusing vaccination in pregnant women, including lack of reliable information, are known. It is unclear what this study adds to existing information in literature, or why this would be different in Poland or Ukraine.

Results for Poland (35,3 %) and Ukraine (16,7 %) are quite different. Why are the results different? What is the cause? I would expect with the same surveys and investigation procedures that results are equal in both groups. This has not been discussed. Are the results significant different? Is the number of inclusions to small? Or are there other reasons? An online study by Skjefte has been discussed (p 8 line 233). The number of inclusions is much higher (2747) than in this study? What does this study add to the literature?

Results are not concise and contains many redundant information. I do not understand why so many characteristics are included in the survey. Table 1 and 2 describe many diseases. I do not understand why these characteristics are analysed. I can invent dozens of diseases more. However, why are these important in the light of the hypotheses of this research? The hypothesis is even not clear.

Discussion: page 7 lines 208 to page 8 line 224 is not discussion on results and belong to introduction or can be even leaved out.

There is a long discussion literature, but the discussion should be restricted to the own results. The discussion should be shortened by 70-80%

Conclusion: page 10 line 323-327. Belong to results.

Conclusion: The level of acceptance ….. is relatively low (line 323-324). How can this be concluded, one should know a reference that stated that de acceptance in other countries is much better. Moreover, in the conclusion it has been expected that the reasons for low acceptance have been described and how that turned out from own research.

It has been concluded that ‘medical consultation concerning the safety of vaccinations against COVID-19 during pregnancy plays an important role in increasing the level of vaccination acceptance among pregnant women’ (conclusion in abstract). I think that this conclusion can be drawn without performing a survey among the pregnant women by simply providing reliable information. What is the added value of the survey? What can be concluded specifically from the survey? That is unclear.

Author Response

Corresponding author at:

Sławomir Januszek, M.D.

Department of Gynecology, Oncology and Obstetrics, Fryderyk Chopin University Hospital No. 1, 35-055 Rzeszów, Poland

Phone No.: +48535860429, e-mail address: [email protected]         

Dear Reviewer,

Thank you very much for your further comments on our manuscript that will surely allow to improve our article. In the attachment, please find corrections in line with your comments and suggestions. We are also sending a revised version of the manuscript. We hope that we have provided adequate explanations to your comments which were used to introduce appropriate corrections.      

           Yours faithfully,                                         

          Sławomir Januszek  

  • The authors describe the role of medical consultation in Poland and Ukraine. The represented group of 300 pregnant women is relatively small and it is unclear whether this group is representative for the whole countries or other countries. This has not been discussed.

The title was modified into: Approach of Pregnant Women from Poland and the Ukraine to Covid-19 Vaccination -The role of medical consultation.

The abstract was modified:

“There are many arguments for the safety and efficacy of COVID-19 vaccines in pregnancy. The aim of the study is to describe the level of vaccination acceptance, to find the factors that most influence the decision to vaccinate, and to describe the scale of changes in vaccination acceptance, influenced by medical information on the safety, efficacy and benefits of vaccination among pregnant women. A total of 300 patients completed the questionnaire, including 150 in Poland and 150 in the Ukraine. The level of vaccination acceptance was assessed before and after medical consultation. There were 53 (35.3%) patients with the intention to get vaccinated in Poland and 25 (16.7%) in the Ukraine. After physician consultation it increased to 109 (72.6%) in Poland and 69 (46%) in the Ukraine. The main factors influencing the acceptance of vaccinations were the fear of harming the foetus (OR-0.119, CI-0.039-0.324 p<0.001), complications in pregnancy (OR-0.073 CI-0.023-0.197 p<0.001) and limitations in the vaccination programme (OR-0.026 CI-0.001-0.207 p<0.001). Medical information about the safety, effectiveness and benefits of vaccinations among pregnant women, provided during a medical visit, may increase the acceptance of vaccinations by 105.6%, as among Polish patients, and even by 176%, as among pregnant women from the Ukraine”.

“The aim of the study is to describe the level of vaccination acceptance in Poland as well as the Ukraine, and to find the factors having the greatest impact on the decision to vaccinate, with particular emphasis on the role of reliable information provided by qualified medical personnel during pregnancy-related medical appointments.” Has been replaced by “The aim of the study is to describe the level of vaccination acceptance and to find the factors that most influence the decision to vaccinate among pregnant women from Poland and Ukraine. Another objective of the research is to describe the scale of changes in vaccination acceptance, influenced by medical information on the safety, efficacy and benefits of vaccination among pregnant women.”in lines 90-94.

“The population of patients enrolled in the study, both in Poland and the Ukraine, is relatively small on the scale of the country and is representative of the population of the studied regions, not the entire country.”-it has been added in lines 250-252.

“In the studied region of Poland, one of the lowest percentages of people vaccinated against COVID-19 is found, while in the Ukrainian population, the problem of vaccination acceptance is quite common [40,41]”. - added in lines 263-265

“Physicians working at clinical centres and patients who come to them for routine check-ups may not completely represent the entire population, in which both knowledge about vaccination and the will to vaccinate may be at different levels. Also, due to the relatively small group of patients under study, in our analysis, we focused on the assessment of indicators and trends that may increase the acceptance of vaccinations”. - modified and supplemented in lines 347-353

  • Pregnant patients who did not consent to the study (page 3, line 116,117) were not included. I suppose that those patients might be against vaccination and might bias the results. Why did they not consent to the survey? This important point has not been discussed.

“Four pregnant women from the Podkarpackie Province and 6 pregnant women from the Chmielnitsky Province did not agree to participate in the study. Patients who do not consent to participation the study may present a low level of acceptance regarding vaccinations against COVID-19. However, due to their small percentage, this should not significantly change the results of the study”.  -  added in lines 246-250

  • There is no comparison with a control group of women that are not pregnant in the same age. How is the vaccination readiness in this group or other groups? Why is the focus (returning the society to the pre-pandemic state page 2 line 72) of the authors on vaccination willingness on pregnant women? The focus of the study is unclear.

“Among non-pregnant women in Poland, in an online study conducted by Babicki et al. on 2,022 respondents, acceptance of vaccinations in the 18-29 and 30-39 age groups in February 2021 was declared by 49.7% and 29.3%, respectively [29]. In an online survey carried out in March 2021 by Sowa et al. [30], among 885 respondents from the general population in Poland, vaccination acceptance totalled 50.8%. In some sources, it is in-dicated that the acceptance of vaccinations among the general Ukrainian population is above 50%, noting, at the same time, a large percentage of health care workers not ac-cepting vaccinations against COVID-19, i.e. 40% [31,32]” - added in lines 256-263.

“It is necessary to understand the factors influencing the acceptance of vaccinations among pregnant women, which will allow to change the level of vaccination acceptance, significantly contributing to the return of society to the pre-pandemic state [16]”. - deleted from lines 74-77.

“What is also worth highlighting is that the COVID-19 pandemic has caused many anxieties among pregnant women about both their own health and the health of the fetus, which significantly affects their well-being [17] ”.- lines 73-76.

“Pregnant women play an important role in receiving the vaccine by other family members”.– has been replaced and supplemented by ”In addition, pregnant women may also play an important role in receiving the vaccine by other family members. This especially applies to vaccination in childhood. In a systematic review of 16 studies, it has been established that attitudes towards pediatric vaccinations indicate, in general, that pregnant women believe vaccines to be important for the protection of their children and the community, but various concerns and misunderstandings persist around vaccine safety and efficacy, which reduce the trust of expectant mothers towards immunization [18]-In the lines 76-82.

  • There is much research on pregnant women in the literature, and factors having the greatest impact for refusing vaccination in pregnant women, including lack of reliable information, are known. It is unclear what this study adds to existing information in literature, or why this would be different in Poland or Ukraine.

“However, the role of this factor has not been fully described” has been replaced by:  “However, in the literature, this is no description regarding a change in the approach to vaccination as a result of medical consultations aimed at providing current knowledge on vaccination safety for the foetus and pregnant woman, including the health benefits of vaccination, vaccine efficacy and current guidelines provided by scientific societies on vaccination”. - in lines 97-101.

“The aim of the study is to describe the level of vaccination acceptance and to find the factors that most influence the decision to vaccinate among pregnant women from Poland and Ukraine. Another objective of the research is to describe the scale of changes in vaccination acceptance, influenced by medical information on the safety, efficacy and benefits of vaccination among pregnant women”. - added in lines 103-107

  • Results for Poland (35,3 %) and Ukraine (16,7 %) are quite different. Why are the results different? What is the cause? I would expect with the same surveys and investigation procedures that results are equal in both groups. This has not been discussed. Are the results significant different? Is the number of inclusions to small? Or are there other reasons? An online study by Skjefte has been discussed (p 8 line 233). The number of inclusions is much higher (2747) than in this study? What does this study add to the literature?

“The difference in the level of vaccination acceptance among Polish and Ukrainian patients before medical consultation was statistically significant at the level of p<0.001. After gynaecological consultations, the number of patients who expressed willingness to undergo vaccination increased to 109 (72.6%) in Poland and 69 (46%) in the Ukraine. The difference in the level of vaccination acceptance among Polish and Ukrainian patients before medical consultation was statistically significant at p<0.001. The increase in the level of acceptance of vaccinations against COVID-19 after a medical consultation was 105.6% in Poland and 176% in Ukraine  - modified in lines 164-172

“Both before and after gynecological consultations among pregnant women from the Ukraine, the level of acceptance and vaccinations was significantly lower, which may be due to many reasons. Although the studied populations did not differ significantly from a medical point of view, the level of education of pregnant women from Poland was higher, and the countries of patient origin differ economically, politically and culturally. The lower acceptance of vaccinations among Ukrainian patients may be due to the non-transparent policy of the Ukrainian authorities regarding the choice of vac-cine manufacturer, the availability of the vaccine, as well as the large number of healthcare professionals who do not accept vaccinations [31]. Other authors point to media disinformation, promotion of the antivaccine movement, political distrust and the issue of purchasing vaccines as the reasons for the low level of acceptance of vac-cination against COVID-19 [32]”.  - added in lines 266-277

“In a study by Skiefte et al., the group of respondents was much larger. In many coun-tries, however, it was an online survey. In our study carried out in a face-to-face manner with doctors, we presented the real extent of change in the approach to vac-cination under the influence of reliable information provided during a medical visit, which was reassessed at a follow-up visit. In our study, we have shown that this one factor can increase the acceptance of vaccinations by 105.6%, as among patients from Poland, or even by 176%, as among pregnant women from the Ukraine.” - modified and supplemented in lines 285-291.

  • Results are not concise and contains many redundant information. I do not understand why so many characteristics are included in the survey. Table 1 and 2 describe many diseases. I do not understand why these characteristics are analysed. I can invent dozens of diseases more. However, why are these important in the light of the hypotheses of this research? The hypothesis is even not clear.

The questionnaire did not include questions about individual diseases. It contained an open-ended question about the diseases the patient suffers from, with a request to mention them or indicate that s/he lacks any diseases.

The diseases mentioned by the patients were analysed in order to show that patients from Poland and the Ukraine do not differ significantly in terms of medicine, and to check whether the general presence of diseases, or whether specific diseases, have significant impact on the acceptance of the vaccine.

Tables 1 and 2 have been removed from the main text and included in the ‘Supplementary materials’.

It was also explained in lines 301-307-“Considering the differences between the Polish and Ukrainian populations, apart from the incidence of psoriasis and allergies, no significant differences were found in comorbidities. Among Polish and Ukrainian patients with hypothyroidism, vaccina-tion against COVID-19 was significantly more often accepted before and following medical consultation. Differences in the level of acceptance among pregnant women with some comorbidities may be related to the perceived higher risk of infection among these patients.”

  • Discussion: page 7 lines 208 to page 8 line 224 is not discussion on results and belong to introduction or can be even leaved out.

This part of text has been removed.

  • There is a long discussion literature, but the discussion should be restricted to the own results. The discussion should be shortened by 70-80%

This part of the text has been shortened significantly.

  • Conclusion: page 10 line 323-327. Belong to results.

This has been removed.

  • Conclusion: The level of acceptance ….. is relatively low (line 323-324). How can this be concluded, one should know a reference that stated that de acceptance in other countries is much better. Moreover, in the conclusion it has been expected that the reasons for low acceptance have been described and how that turned out from own research.

In lines 277-280 studies are listed in which the acceptance of vaccinations by pregnant women was significantly higher.

“In a study conducted by Skjefte et al. [1] among pregnant women in 19 countries, 52.0% (n = 2747) declared the intention to undergo vaccination for COVID-19, while the highest acceptance levels were noted in a study by Tao et al. [33] carried out in China (77.4%), by Mohan et al. [25] in Qatar (75%), and in Italy (74.5%) in a research trial carried out by Mappa et al. [26].

Reasons for the low acceptance has been supplemented in lines 351-354 -“Concerns about vaccine-related side effects or complications, harm to the foetus, and limitations of the vaccination programme were significant reasons for the low level of vaccination acceptance and important predictors of lower vaccination likelihood”.

  • It has been concluded that ‘medical consultation concerning the safety of vaccinations against COVID-19 during pregnancy plays an important role in increasing the level of vaccination acceptance among pregnant women’ (conclusion in abstract). I think that this conclusion can be drawn without performing a survey among the pregnant women by simply providing reliable information. What is the added value of the survey? What can be concluded specifically from the survey? That is unclear.

“Medical information about the safety, effectiveness and benefits of vaccinations among pregnant women, provided during a medical visit, may increase the acceptance of vaccinations by 105.6%, as among Polish patients, and even by 176%, as among pregnant women from the Ukraine”. - modified in lines 354-357.

Reviewer 2 Report

Thank you for the opportunity to revise this manuscript. The authors present interesting research about a topical issue. Vaccine hesitancy is a major threat to achieving herd immunity across different populations and describing various scenarios can be useful to detect cultural barriers.

However, I suggest some improvements.

Abstract. This section should follow the format requirements of the journal. No sub-headings.

The introduction could be expanded with experiences from other countries that investigated a similar topic before the pandemic both in relation to acceptance of a vaccine from the mother but also for her children. Good references worth including are:

  • Danchin MH, et al. Vaccine decision-making begins in pregnancy: Correlation between vaccine concerns, intentions and maternal vaccination with subsequent childhood vaccine uptake. Vaccine, 2018. 36(44), 6473-6479.
  • Samannodi M, et al. COVID-19 Vaccine Acceptability Among Women Who are Pregnant or Planning for Pregnancy in Saudi Arabia: A Cross-Sectional Study. Patient Prefer Adherence. 2021 Nov 23;15:2609-2618.
  • Rosso A, et al. Factors affecting the vaccination choices of pregnant women for their children: a systematic review of the literature. Human vaccines & immunotherapeutic. 2020;16(8):1969-1980.

Methods. Search strategy is not applicable to this study.

Also, this is a before-after design… defining it as prospective does not mean anything.

Multivariate or multivariable analyses?

Results. The response rate should be reported. In case of low response rate, the potential selection bias should be discussed.

Among the modifiable factors that can influence vaccine acceptance, health literacy plays a key role. A recent meta-analysis found that the levels of literacy are generally quite low in Europe. This should be added to the discussion (reference: Baccolini V, et al. What is the Prevalence of Low Health Literacy in European Union Member States? A Systematic Review and Meta-analysis. J Gen Intern Med. 2021;36(3):753-761.)

It is also necessary to specify what is meant by “medical consultant”. It is the usual gynecologist? Is it the general practitioner? Is it the vaccinating physician?

It is necessary to clarify how confidence in the health care provider proposing vaccination was determined

Author Response

Corresponding author at:

Sławomir Januszek, M.D.

Department of Gynecology, Oncology and Obstetrics, Fryderyk Chopin University Hospital No. 1, 35-055 Rzeszów, Poland

Phone: +48535860429, e-mail address: [email protected]         

Dear Reviewer,

Thank you very much for your further comments on our manuscript that will surely allow for the improvement of our article. Please find our corrections in line with your comments and suggestions. We are also sending a revised version of the manuscript. We hope that we have provided extensive explanations to your comments, which were used to introduce appropriate corrections.      

           Yours faithfully,                                         

          Sławomir Januszek

Thank you for the opportunity to revise this manuscript. The authors present interesting research about a topical issue. Vaccine hesitancy is a major threat to achieving herd immunity across different populations and describing various scenarios can be useful to detect cultural barriers.

Changes to the text are highlighted in yellow.

However, I suggest some improvements.

  • This section should follow the format requirements of the journal. No sub-headings.

   Thank you for your comments - we have changed the ‘Abstract’ section according your suggestions.

  • The introduction could be expanded with experiences from other countries that investigated a similar topic before the pandemic both in relation to acceptance of a vaccine from the mother but also for her children. Good references worth including are: Danchin MH, et al. Vaccine decision-making begins in pregnancy: Correlation between vaccine concerns, intentions and maternal vaccination with subsequent childhood vaccine uptake. Vaccine, 2018. 36(44), 6473-6479.

 Samannodi M, et al. COVID-19 Vaccine Acceptability Among Women Who are Pregnant or Planning for Pregnancy in Saudi Arabia: A Cross-Sectional Study. Patient Prefer Adherence. 2021 Nov 23;15:2609-2618.

Rosso A, et al. Factors affecting the vaccination choices of pregnant women for their children: a systematic review of the literature. Human vaccines & immunotherapeutic. 2020;16(8):1969-1980.

-Thank you for your suggestion - we have added the appropriate indicated references and extended the text in 2 paragraphs: ‘Introduction’ and ‘Discussion’ sections.

3) Methods. Search strategy is not applicable to this study.

-Thank you - we have changed to the name of this section to ‘Material and methods’.

4) Also, this is a before-after design… defining it as prospective does not mean anything.

-Thank you - we have explained that our project was a cross-sectional study.

5) Multivariate or multivariable analyses?

This has been consistently changed to ‘multivariate’ analysis in the MSC.

7) Results. The response rate should be reported. In case of low response rate, the potential selection bias should be discussed.

It has been suplemented in paragraph 3.1 (lines 160-161)

The respondents answered a total of 96% of the questions, while the range of answers to individual survey questions from 1 to 47 ranged from 94-100%.

8) Among the modifiable factors that can influence vaccine acceptance, health literacy plays a key role. A recent meta-analysis found that the levels of literacy are generally quite low in Europe. This should be added to the discussion (reference: Baccolini V, et al. What is the Prevalence of Low Health Literacy in European Union Member States? A Systematic Review and Meta-analysis. J Gen Intern Med. 2021;36(3):753-761.)

-Thank you for your suggestion - we have added the appropriate significant results of this meta-analysis to paragraph 4.1 in the ‘Discussion’ section.

9) It is also necessary to specify what is meant by “medical consultant”. It is the usual gynecologist? Is it the general practitioner? Is it the vaccinating physician?

The medical consultation was carried out by a physician in the course of specialization in gynecology or by a specialist in gynecology. We have explained it in the lines 117-119.

10) It is necessary to clarify how confidence in the health care provider proposing vaccination was determined

- Thank you - we have added an explanation in paragraph 2.1. (lines 117-123)

The visits to the gynecologist were planned as part of a pregnancy health check-up. Each woman had the opportunity to individually choose the gynecologist who was to monitor their pregnancy.

Round 2

Reviewer 1 Report

Among the population of pregnant women there are so many women with a disease (table 1S, more that 30% of the women). This surprises me. Do pregnant women without an additional disease not visit a gynaecologist? Is the population that is included for this research representative for the pregnant population? Perhaps an ‘unhealthy’ population of pregnant woman could accept vaccination differently than population of pregnant woman without underlying disease. Please make this clear.

The design of Table 1 is not clear. It describes the predictors of vaccinations. Not only positive predictors (predictors for accepting vaccination) but also negative predictors (predictors for refusing vaccinations) before and after visiting the gynaecologist. It shows which questions of the survey are important, however I think it is important to show odds ratios before and after visiting a gynaecologist. A redesign of the table by changing the rows ‘before visiting a gynaecologist’ and ‘after visiting a gynaecologist’ to a column in the table and showing how odds ratios changes from before to after would make the table more clear.

Figure 2 is unclear for me. What does it tell? Is this necessary for the reader or can it leaved out.

Table 2 is unclear for me, I cannot follow message. I cannot see the relation between before and after medical visit in the rows and columns. Is it possible to reduce the table (leave column% and row%)

Conclusion/title: The level of acceptance …In Poland and Ukraine... is relatively low (line 344-345). De authors state in their limitations that the invested population may not be representative for the whole countries. Local aspects, like social-economic aspects and education level may be different as stated. The authors should be more prudent in their conclusion.

Author Response

Corresponding author at:

Sławomir Januszek, M.D.

Department of Gynecology, Oncology and Obstetrics, Fryderyk Chopin University Hospital No. 1, 35-055 Rzeszów, Poland

Phone No.: +48535860429, e-mail address: [email protected]         

Dear Reviewer,

 Thank you very much for your further comments on our manuscript, which will certainly help improve our article. Please find enclosed corrections in line with your comments and suggestions. We have also attached a revised version of the manuscript. We hope that the introduced corrections will enable the publication of this article.

          Yours faithfully,                                         

          Sławomir Januszek  

  • Among the population of pregnant women there are so many women with a disease (table 1S, more that 30% of the women). This surprises me. Do pregnant women without an additional disease not visit a gynaecologist? Is the population that is included for this research representative for the pregnant population? Perhaps an ‘unhealthy’ population of pregnant woman could accept vaccination differently than population of pregnant woman without underlying disease. Please make this clear.

“In the study group, 16.8% of patients mentioned comorbidities, and 82.4% reported more than 3 diseases (e.g. obesity, diabetes, hypertension, hypothyroidism in the same patient).” It was added in the lines-175-177

“The proportion of patients with comorbidities was quite high, which may indicate that the selected population is not representative of the pregnant population. However, patients reported comorbidities in the questionnaire by themselves (the questionnaire was anonymous). Some of the diseases mentioned by pregnant women are seasonal allergies or misinterpreted obesity in pregnancy. Many diseases reported by outpatients as disorders could only be part of the diagnostic process, such as blood pressure or glucose monitoring.” It was added in lines 301-307.

Issues in the discussion section related to the results were removed:

“Also, patients with hypothyroidism significantly more often accepted vaccination against COVID-19 before and after consultation” -it was deleted from lines 298-299

“Also, patients with hypothyroidism significantly more often accepted vaccination against COVID-19 before and after consultation” -it was deleted from lines 307-309.

“Among Polish and Ukrainian patients with hypothyroidism, vaccination against COVID-19 was significantly more often accepted before and following medical con-sultation. Differences in the level of acceptance among pregnant women with some comorbidities may be related to the perceived higher risk of infection among these pa-tients.” It was modified into “Among Polish and Ukrainian patients with hypothyroidism, vaccination against COVID-19 was significantly more commonly accepted, which may be related to a higher perceived risk of infection.” In lines 307-309.

“Patients choosing more experienced centers for pregnancy control visits may have a greater burden of comorbidities, and their level of acceptance may be different than in the entire population of pregnant women.”- it was added in lines 350-352.

  • The design of Table 1 is not clear. It describes the predictors of vaccinations. Not only positive predictors (predictors for accepting vaccination) but also negative predictors (predictors for refusing vaccinations) before and after visiting the gynaecologist. It shows which questions of the survey are important, however I think it is important to show odds ratios before and after visiting a gynaecologist. A redesign of the table by changing the rows ‘before visiting a gynaecologist’ and ‘after visiting a gynaecologist’ to a column in the table and showing how odds ratios changes from before to after would make the table more clear.

The table was redesigned.

  • Figure 2 is unclear for me. What does it tell? Is this necessary for the reader or can it leaved out.

The Figure 2. was removed.

  • Table 2 is unclear for me, I cannot follow message. I cannot see the relation between before and after medical visit in the rows and columns. Is it possible to reduce the table (leave column% and row%)

Table 2 was modified.

  • Conclusion/title: The level of acceptance …In Poland and Ukraine... is relatively low (line 344-345). De authors state in their limitations that the invested population may not be representative for the whole countries. Local aspects, like social-economic aspects and education level may be different as stated. The authors should be more prudent in their conclusion.

“The level of acceptance regarding vaccination against COVID-19 among pregnant patients in Poland (35.3%) and the Ukraine (16.7%) is relatively low.” Was replaced by “The level of acceptance of vaccinations against COVID-19 in the studied group of pregnant women from Poland (35.3%) and Ukraine (16.7%) was relatively low.” In lines 357-358.
